# 3-D Image-Driven Morphological Crop Analysis: A Novel Method for Detection of Sunflower Broomrape Initial Subsoil Parasitism

**DOI:** 10.3390/s19071569

**Published:** 2019-04-01

**Authors:** Ran Nisim Lati, Sagi Filin, Bashar Elnashef, Hanan Eizenberg

**Affiliations:** 1Department of Plant Pathology and Weed Research, Newe Ya’ar Research Center, Agricultural Research Organization, Ramat Yishay 30095, Israel; eizenber@volcani.agri.gov.il; 2Mapping and Geo-Information Engineering, Technion-Israel Institute of Technology, Haifa 3200003, Israel; filin@technion.ac.il (S.F.); bashar.hz@gmail.com (B.E.)

**Keywords:** multi-view stereo, plant phenotyping, segmentation

## Abstract

Effective control of the parasitic weed sunflower broomrape (*Orobanche cumana* Wallr.) can be achieved by herbicides application in early parasitism stages. However, the growing environmental concerns associated with herbicide treatments have motivated the adoption of precise chemical control approaches that detect and treat infested areas exclusively. The main challenge in developing such control practices for *O. cumana* lies in the fact that most of its life-cycle occurs in the soil sub-surface and by the time shoots emerge and become observable, the damage to the crop is irreversible. This paper approaches early *O. cumana* detection by hypothesizing that its parasitism already impacts the host plant morphology at the sub-soil surface developmental stage. To validate this hypothesis, *O. cumana-* infested sunflower and non-infested control plants were grown in pots and imaged weekly over 45-day period. Three-dimensional plant models were reconstructed using image-based multi-view stereo followed by derivation of their morphological parameters, down to the organ-level. Among the parameters estimated, height and first internode length were the earliest definitive indicators of infection. Furthermore, the detection timing of both parameters was early enough for herbicide post-emergence application. Considering the fact that 3-D morphological modeling is nondestructive, is based on commercially available RGB sensors and can be used under natural illumination; this approach holds potential contribution for site specific pre-emergence managements of parasitic weeds and as a phenotyping tool in *O. cumana* resistant sunflower breeding projects.

## 1. Introduction 

Sunflower broomrape (*Orobanche cumana* Wallr.) is a chlorophyll-lacking holoparasitic weed, which is considered a major biotic factor limiting sunflower production in large areas of Europe and Asia [1]. The main growth stages of this root parasite include dormancy, seed germination, attachment to the host root, connection to the host tissues, and tubercle production [1]. *Orobanche cumana* growth depends completely on the host plant and, once attached to the root system of the plant, the weed depletes the host nutrients, minerals, and water, and may cause severe damage to the yield quality and quantity [2]. The most significant stages of the *O. cumana* life cycle, in terms of plant pathogenicity, occur before the weed emerges above the surface of the soil. By the time the inflorescences become observable, the damage to the crop has already been done [3].

Adequate control of *O. cumana* is achieved mainly by herbicide application [4]. Low rates of imazapic applied at a specific developmental stage of *O. cumana* tubercle (which can be predicted by growth models) have proved effective in controlling this weed [5,6]. However, the current herbicide-based management ignores spatial variations within and between fields, leading to homogeneous application over the entire area. It is thus likely that large amounts of herbicides are being used for no reason [7,8] and that targeting herbicide application to broomrape-infected areas exclusively would markedly reduce herbicide usage [9]. Nonetheless, since the critical growth stages of this parasite occur underneath the soil surface, available sensing methods cannot indicate the presence of the parasitic weed [7]. 

It has been hypothesized that spectral sensing can be employed to monitor *O. cumana* establishment, as the host plant may be under water stress, which in turn, may be reflected by different absorbance patterns [7]. Recently, several groups used hyperspectral and fluorescence imaging for early detection of *O. cumana* parasitism in sunflower by characterizing different reflectance patterns in *O. cumana* infested as compared to non-infested control plants [10,11]. Cochavi et al. (2017) showed that *O. cumana* parasitism initially affects the nutrient content within the leaf, eliciting changes to the structure of the mesophyll tissue. A far-NIR (800–1300 nm) and SWIR (1450 nm range) reflectance values were then found to be correlative with this structural change, and were proposed as an indicator of early *O. cumana* parasitism when the parasite is attached to the host root. Ortiz-Bustos et al. (2016) observed changes in the leaf chlorophyll content due to *O. cumana* infestation that was detected via differences in the reflectance pattern at the 680 nm and 740 nm wavelengths. However, the Ortiz-Bustos’ group imaged plants under controlled laboratory conditions and applied excitation of a specific wavelength during imaging (UV light at 360 nm). These conditions may not be translatable to natural light conditions, where reflectance patterns of infected plants and the corresponding detection abilities of hyperspectral imaging may be less robust. These limitations emphasize the need to evaluate an alternative detection approach that may be more effective under natural light conditions. 

With advances in modeling algorithms and increased computational power, 3-D plant reconstruction has become a viable task, enabling the generation of models with higher resolution and better description of the plants themselves and of their canopy architecture [12,13]. Furthermore, 3-D reconstruction techniques allow for autonomous segmentation of the plant point cloud into leaf and stem-related points and the extraction of morphological parameters, down to the single organ level [12,14,15]. Such detailed analysis of plant morphology can be employed for various agronomic purposes [13,16,17,18], including weed detection [19]. In this context, water and nutrient uptake due to *O. cumana* attachment may inhibit the sunflower plant growth. Nonetheless, a potential relationship between *O. cumana* parasitism and the host plant morphology has not been studied. Thus, the current work set out to assess the possible use of 3-D image-driven models for early detection of *O. cumana* parasitism by analyzing morphological variations in the host sunflower plant.

## 2. Materials and Methods

Experiments were conducted in ten (10 L) pots that were placed in a net-house. Half of the pots were filled with soil containing 10 parts per million (ppm) of *O. cumana* seeds that were mixed with the soil to homogeneity. The other half was not mixed *O. cumana* seeds and served as control. A single sunflower plant was planted and grown in each pot. Beginning 16 days after planting (DAP), the plants were imaged at 7-day intervals, for 4 weeks, corresponding to 45 DAP.

### 2.1. Image Acquisition and Analysis

The imaging and data analysis steps are described in Figure 1. Images were acquired, under natural illumination, in the net house with a Nikon D90 camera fitted with a Nikkor 24 mm lens. At each imaging session, 20 images were captured, irrespective of growth stage, in a hemispherical coverage, from a ~1 m distance (Figure 2). The overlapping area between consecutive images was approximately 65%. A structure from motion (SfM) multi-view stereo (MVS) (using Pix4DMapper photogrammetric software) was used for generating 3-D point clouds of the plants. In general, the photogrammetric process involves detection and matching of corresponding image keypoints and estimation of the image orientations. The orientation stage is based on a bundle adjustment model that simultaneously computes the image orientation and estimates the keypoint positions in 3-D [20]. Because the reconstructed point clouds cover the entire scene (Figure 2), prior to the derivation and evaluation of the morphological parameters, a pre-processing stage was applied, which consisted of scaling and cropping the plant-related points.

### 2.2. Plant Segmentation and Parameter Extraction

Four growth parameters were chosen to evaluate the plant morphology: height, volume, width, and first internode length of the plant. All four were autonomously extracted from the 3-D point cloud. Height, defined by the vertical distance between the soil surface and the uppermost point, was measured by the difference between the minimal and maximal *z* ordinates. Plant volume, which is known to strongly correlate with the shoot dry weight, a fundamental growth indicator, was computed using spatial partitioning into volumetric cells (voxels). The conversion was based on the approach described by reference [21]. Summing all the voxels containing reconstructed plant-related points yielded the volume. Sets of connected voxels that were smaller than five cells were considered as noise and were discarded. Plant width, a 2-D parameter, was obtained by projecting the plant’s 3-D points onto the *x*-*y* plane and computing the maximal planar difference.

The internode length is a single-organ morphological parameter, as opposed to height, width and volume, which relate to the whole plant. To extract this parameter, the reconstructed data were first segmented into leaf- and stem-related points (Figure 3). For this purpose, a 3-D tensor-driven approach, which uses first- and second-order tensor information, was used [22]. A first-order tensor **t***_i_* at point **x***_i_*, is the sum of vectors between a point and its neighbor points:(1)$$t_{i}={\left[\begin{array}{ccc} t_{x} & t_{y} & t_{z} \end{array}\right]}^{T}=\sum_{j\in \;\sigma }^{}\left(x_{i}-x_{j}\right),$$
with *σ* being the neighborhood of **x***_i_*. The first-order tensor facilitates the evaluation of the distribution of points around **x***_i_*. A second-order tensor is the sum of the outer products:(2)$$T_{i}=\sum_{j\in \;\sigma_{i}}^{}\left(x_{j}-x_{i}\right){\left(x_{j}-x_{i}\right)}^{T}=\left[\begin{array}{ccc} t_{xx} & t_{xy} & t_{xz}\\ & t_{yy} & t_{yz}\\ Sym. & & t_{zz} \end{array}\right],$$
where **T***_i_* is the second-order tensor at point *i*, **x**_*i*_ is the point for which the tensor is computed, and **x**_*j*_ is a point in its neighborhood, *σ*. The outer product form Equation (1) establishes a symmetric positive-semidefinite matrix whose spectral decomposition into its eigenvectors and eigenvalues can be written as:(3)$$T_{3\times 3}=V\Lambda V^{T}=\sum_{i=1}^{n}\lambda_{i}v_{i}v_{i}^{T},$$
with **Λ** = *diag*(*λ*_1_
*λ*_2_
*λ*_3_), a non-negative diagonal matrix composed of its eigenvalues, and **V** = [**v**_1_
**v**_2_
**v**_3_], an orthonormal matrix formed by concatenation of the eigenvectors. Because **V** preserves lengths and angles (isometry), it is referred to as the tensor orientation, while the *λ_i_* terms are the strengths of the respective orientation. Equation (3) can be rearranged as: (4)$$T_{3\times 3}=(\lambda_{1}-\lambda_{2})v_{1}v_{1}^{T}+(\lambda_{2}-\lambda_{3})(v_{1}v_{1}^{T}+v_{2}v_{2}^{T})+\lambda_{3}(v_{1}v_{1}^{T}+v_{2}v_{2}^{T}+v_{3}v_{3}^{T})$$

The geometrical interpretation of Equation (4) can be visualized in Figure 4, which shows how the second tensor representation can decompose to linear, planar and spherical components, with the eigenvalue relations dictating their significance. For instance, the axial distribution of a point would result in eigenvalues as follows: *λ*_1_ > 0, and *λ*_2_ = *λ*_3_ = 0. In a surface-like distribution, *λ*_1_, *λ*_2_ > 0, while *λ*_3_ = 0. The differences between the first two eigenvalues in such an arrangement provide information about the uniformity of the point distribution. The third eigenvector *λ*_3_ reflects the deviation from planarity, and analysis shows that in such a distribution, this value rarely exceeds 10% of the magnitude of the first eigenvector. Therefore, it is possible to perform a classification that is based on a straightforward analysis of the first two eigenvalues, while ensuring that the third one is significantly smaller. For the leaf parts that can be related as surface-like features, points should be evenly distributed on the dominant plane-of-projection, and therefore, we assume *λ*_1_ ≅ *λ*_2_. For stem-related points, a noticeable variation between the first and second eigenvalues is expected, which indicates an elongated axial form.

Our experiments showed that when basing exclusively on the second-order tensor-based classification, successful classification rates of 85%–93% can be achieved, with most misclassifications observed for boundary points, such as leaf edges and voids, in the reconstruction. Misclassifications are tackled by utilizing the information encapsulated in the first-order tensor (Equation (1)). As mentioned, the first-order tensor holds information about the neighborhood arrangement about xi; both direction and magnitude can be extracted from it. When the neighborhood arrangement is nearly symmetrical (typically for the leaf area), the magnitude of the first-order tensor would be close to zero, and is expected to grow as the asymmetry of distribution increases. For that reason, points located on leaf boundaries are expected to have high values of magnitude in contrast to points on the center of the leaf, and therefore, misclassification in leaf-related areas can be expected. In contrast, because of the symmetrical elongated form of stems, the magnitude of points, even if located on the edges, will tend to be small.

Following the segmentation phase (Figure 5B) and extraction of the stem-related points (Figure 5C), the location of the internodes along the stem are extracted. Unlike standard stem points, the nodes are the meeting points between the stem and petioles, and therefore have a broader spatial point distribution. In consequence, the second-order tensor at these locations has larger a *λ*_2_ value as compared to the standard stem points (Figure 5D). To define the location of the internodes, we first detected regions along the stem with high *λ*_2_ value and then computed the centroid of this set of points and considered it as the internode location.

### 2.3. Minirhizotron Experiments

Implementation of our detection approach for field application requires monitoring infected sunflower plants during the period during when herbicide treatment is effective. Therefore, another set of experiments was performed to validate whether the detection timing of *O. cumana* parasitism lies within the critical time frame for herbicide application, as was done in previous studies [6]. *O.-cumana*-infested sunflowers were grown at the same time and location as those grown under controlled conditions (Section 2). The plants were grown in different pots from those previously described, in order to accommodate the minirhizotron imaging system used for in situ monitoring of the underground developmental stages of the parasite [23]. The system and setups were similar to those described in reference [23]. The number of germinated *O. cumana* seeds and attachments, their penetration into the sunflower roots, and tubercle size and number were extracted from images acquired by the system. Additionally, temperature was recorded hourly throughout the study, with data loggers located 5 cm below the soil surface and converted to thermal-time units, i.e., growing degree days (GDD), rather than chronological time [24]. The phenology stage of the plant at each imaging time was evaluated using the conventional phenological stage methodology, as described in reference [25].

### 2.4. Statistical Analysis

The effect of *O. cumana* infestation was studied using a completely randomized experimental design. Infestation was tested at the single level factor (infestation [10 ppm] vs. non-infestation [control]), with five replications (plants). The experimental unit was a single sunflower plant from which the 3-D model was reconstructed and parameters were extracted. A separate parameter analysis was performed for all plants following each imaging session. The estimated values for sunflower height, width, volume and first internode length were subjected to ANOVA, and means were compared using the Tukey-HSD test (*p* ≤ 0.05). 

Since our study was one of the first to extract and estimate internode length, an analysis of the accuracy of the measurement of this organ-level parameter was performed. The estimated and manually measured first internode length values were regressed using a linear model, and the root mean square error (RMSE) was evaluated. Then, the percentage error was estimated as the ratio between the computed RMSE value and the average actual internode length value measured in this study. For the minirhizotron experiment, the number of *O. cumana* attachments from all sessions was non-linearly regressed against GDD, as described by reference [23].

## 3. Results

### 3.1. O. Cumana Parasitism Dynamics: Minirhizotron Experiments

Parasitism stages of *O. cumana* observed by the minirhizotron system included attachments, development of tubercles of different sizes and meristem initiation (Figure 6). The repeated measurements at 550 GDD (V 2–4), 750 GDD (V 6–8), 940 GDD (V 10–12) and 1120 GDD (R1), provided an indication of the increase in the number of tubercle attachments between evaluations, and the regression analysis using the logistic equation yielded a strong relationship (*R*^2^ = 0.98) between the number of *O. cumana* attachments and thermal time (Figure 7 and Table 1). The parasitism pattern of *O. cumana* on sunflower was similar to that observed by other researchers [2,6].

### 3.2. 3-D Reconstruction and Internode Estimation 

The 3-D model provided a detailed reconstruction (Figure 2 and Figure 3) of two young, 15-cm-tall plants at 550 GDD (16 DAP), and well-developed as >1 m-tall plants with a large number of fully developed leaves, at 1120 GDD (45 DAP). Our point cloud segmentation model distinguished well between stem- and leaf-related points (Figure 3 and Figure 5). Of note, the stems and the petioles were separated from the leaf-related points, irrespective of the developmental stage of the plants (Figure 5). Based on the extraction of the stem-related points, the exact petiole connection points to the stems were detected (Figure 8), which enabled autonomous estimation of the length between internodes. The manually measured first internode lengths exhibited a strong linear correlation with the estimated values (R^2^ = 0.99, slope (*a*) = 1.02 (Figure 9)). Furthermore, the RMSE value was below 7 mm, which was ~3.5% error, indicating that the segmentation model provided accurate morphological estimates at the single-organ level.

### 3.3. Morphological Analysis for O. Cumana Detection

Comparative results for the morphological parameters of control vs. infected plants are plotted in Figure 10. Statistically significant differences between control and infected groups were observed for plant height, internode length and volume, with level of significance and the earliest time at which differences were observed varying with the parameter (Figure 10). Plant height and the first internode length provided a significant early morphological indication of broomrape infection (Figure 10A,B), while significant differences in the volume parameter were only registered towards the end of the GDD period studied (Figure 10C). 

Already at 750 GDD, a significant difference of 7 cm, equivalent to 17% difference, was evident between control and infected plant heights (Figure 10A). The difference increased from there on, reaching 30 cm (equivalent to 36%) at 1120 GDD, 112 and 82 cm for the control and infected plants, respectively (Figure 10A). The first internode length exhibited an even stronger response to broomrape infection, and at 750 GDD, a significant length difference of 23 cm vs. 13 cm (amounting to 76% difference) for the control and infected plants, respectively, was recorded. As with plant height, this trend continued, and at the final evaluation (1120 GDD), the first internode differences had reached 106%, namely 31 cm vs. 15 cm, for control and infected plants, respectively (Figure 10B). In contrast to the height and internode length measures, similar volumes were estimated at the evaluations at 550 and 750 GDD (Figure 10C). A significant difference in volume (40%) was observed at 940 GDD, namely, 0.2 and 0.12 m^3^ for the control and infected plants, respectively. The volume difference between infected and control plants was the highest at the end of the study, 125% (Figure 10C). 

Plant width was the only morphological parameter extractable using monoscopic 2-D imaging, without the need for 3-D shape reconstruction. Estimations of this parameter ranged from 20 cm at the first evaluation to 66 cm at the last, but no significant difference (*p* > 0.093) was observed between infected and control plants throughout the study (Figure 10D). For example, in the third evaluation (940 GDD), when the other parameters showed significant differences (>27% difference), the estimated width values were 55 cm and 61 cm in control and infected plants, respectively, with a nonsignificant 9% difference (Figure 10D). In summary, the three 3-D related parameters clearly showed the impact of *O*. *cumana* infection on plant morphology, while the 2-D parameter, width, was not a suitable parameter for predicting *O. cumana* infection.

## 4. Discussion

This study presents an image-driven approach for monitoring early parasitism of *O. cumana* on sunflower roots. The study revealed the relationships between *O. cumana* parasitism and host plant 3-D morphological growth parameters. Compared to non *O. cumana*- infested control plants, infected plants were significantly shorter and had shorter internode lengths and lower plant volumes (Figure 10A–C, respectively). Furthermore, the times at which these morphological differences became observable were early enough for herbicide application before irreversible damage is caused to the sunflower plants. Both plant height and first internode length showed significant differences as early as 750 GDD (Figure 10A,B), a phenological stage (V 6–8) that is relevant for effective post-attachments imazapic application in confectionary sunflower that can still prevent *O. cumana* damage [2,6]. To the best of our knowledge, this is the first study to provide quantitative evidence of the impact of broomrape parasitism on early morphological features in host plant.

This study was one of the first to use imaging as a non-destructive detection tool for broomrape establishment. Previous studies that tested hyperspectral imaging for the same purpose, found the reflectance values of the far-NIR (800–1300 nm) and the SWIR (1450 nm range) wavelengths to be indicative of broomrape infection [10,11]. The measured value at these wavelengths was 1% and 1.5% lower, respectively, in broomrape infected plants compared to non-infested control. However, in comparison to the 17% height differences observed in our study, the 1.5% reflectance differences are slim and may not be sufficiently robust for detection under varying field illuminations. Furthermore, the significant differences in the reflectance values were only observed after 31 days, a time that may not be relevant for herbicide application. On days 36 and 51 after planting, only a slight increase in the reflectance difference between infected and control plants was observed, whereas morphological differences detected by the 3-D model continued to increase through the course of the 45 days study. The low cost and high viability of the sensors needed for 3-D modeling is another advantage of morphological assessments over hyperspectral imaging [26]. 

Another contribution of this study lies in our segmentation approach and its ability to estimate node location and the distance between nodes. The internode-length is known to be affected by various biotic (e.g., soil-borne pathogens) and abiotic (e.g., temperature, water shortage) stresses [27,28]. Therefore, autonomous and non-destructive tools for estimating its value are of great importance for breeders and agronomists [29]. Other 3-D modeling-driven studies have indirectly estimated the internode length by projecting the leaf centers onto the plant stem and assuming this distance to be similar to the internode length [17]. However, the accuracy of this method can be influenced by the growth status of the plant, such as its exposure to stress. In contrast, the model described in this paper estimates the internode length directly (Figure 9). Alternative analysis methods such as skeletonization and local feature descriptors, which are based on 2-D geometric features, have previously been used for internode detection [29,30], but such approaches ignore the actual shape of the plant and its leaf-related parameters.

As our results show, 3-D models can provide continuous and non-destructive phenotypic estimations of plants, starting from early growth stages and continuing through the entire course of the growing season. Thus, using 3-D models for breeding projects can provide early and in-depth insight into the dynamics of broomrape establishment and can save breeders time, labor and resources in the process of selection of resistant lines [4,5,31,32]. Furthermore, our phenotyping approach is based on readily available consumer-grade cameras for documentation, compared to other 3-D sensors, such as light detection and ranging (LiDAR) scanners [33,34,35] or ultrasonic detectors [36,37]. In terms of organ-level phenotyping, Paulus et al. (2014) stated that any pipeline that aims for high-throughput applications must have automated organ separation and data analysis abilities [38]. The authors noted that segmentation methods for 3-D plant models are scarce, and most related analyses are performed manually or require long computation times [39,40]. Here, a time effective model that is based on local computation of characteristic features and that does not involve pre-defined constraints or explicit plant geometry modeling is presented.

3-D modeling has proven to be applicable for a variety of crop species [12,13,14,38,41]. Thus, the proposed early detection approach may prove suitable for other broomrape-infected crops with effective post-emergence (POST) application, expanding the potential impact of the use of these systems [1]. Even though other biotic/abiotic factors may impact the host plant morphology, a certain degree of false detection of the broomrape infection is still useful compared to the current situation, which lacks tools for early detection. Furthermore, combining our 3-D-based detection approach with field history records, pre-season mapping of broomrape patches [42], and temporal parasitism dynamics model [3,7] can reduce the risk of false detection under actual field conditions

Using our 3-D modeling approach for field-scale work will require further research. However, it can be assumed that it will be most effective with single-sided (180°) images acquired from a close range, with cameras mounted on tractors or other mobile platforms. Close-range, single-sided imaging has already demonstrated accurate phenotyping abilities [22], and relevance for various precision agricultural purposes, including monitoring of crop height and biomass under actual field conditions [43,44]. Overlapping canopies of adjacent crop plants, compared to single plants that were used in the current study, will likely have non-significant impact on the efficacy of our modeling approach. Plant height and biomass can be extracted from well-developed plants [45], while stem (internode-length) reconstruction might require viewing imaging point adjustment [46]. Furthermore, Nakarmi and Tang (2012) used 3-D data acquired from a single-sided viewpoint to identify corn stems to set their location and evaluate distance between plants in the field [47], suggesting that our organ level analysis and internode length extraction can be applied in the field.

## 5. Conclusions

Effective management of *O. cumana* requires herbicide application at early stages of parasitism, when no definite above-ground indications are detectable. This work demonstrated that *O. cumana* establishment on sunflower is associated with variations in host morphology that can be monitored in early parasitism stages using 3-D shape analysis of the host. Furthermore, we have shown that nodes and internode distances can be accurately estimated, and contribute greatly to the detection ability. Future work will focus on how this spatial model can be used in the field in real-time applications combined with a parasitism dynamics temporal model. 

## Figures and Tables

**Figure 1 sensors-19-01569-f001:**
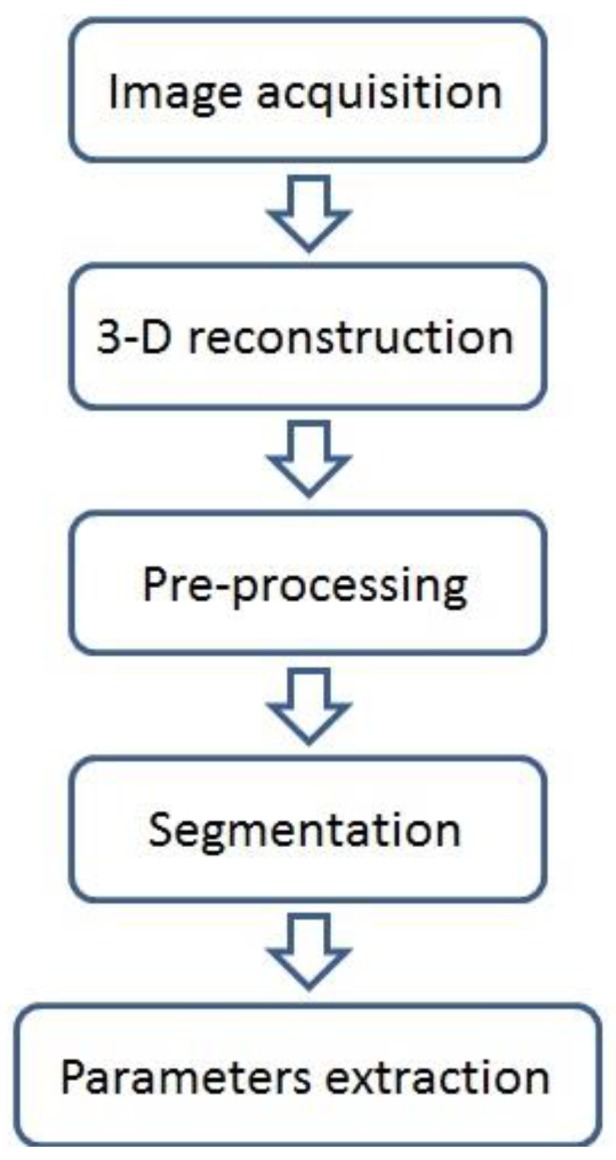
Flowchart of image data processing for 3-D-based morphological parameter extraction.

**Figure 2 sensors-19-01569-f002:**
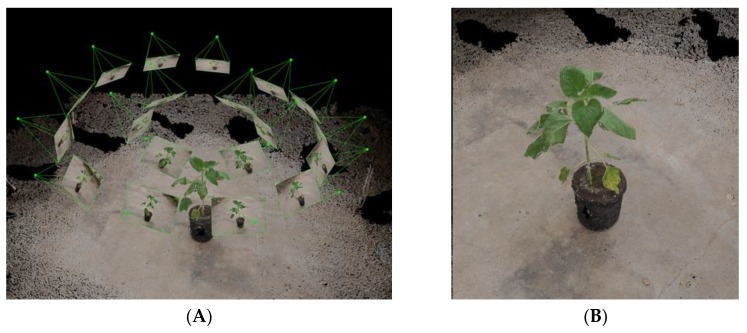
Imaging setup and 3-D reconstruction. Camera positions (‘pyramid’ apices), orientations and imaged areas of the acquired data (**A**) and the 3-D reconstructed model of the entire scene before the pre-processing stage (**B**).

**Figure 3 sensors-19-01569-f003:**
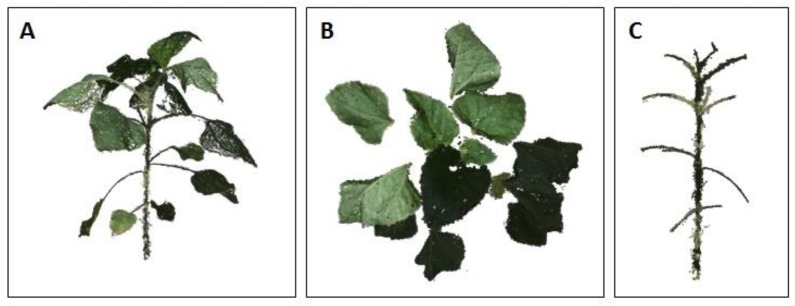
Plant segmentation. The reconstructed 3-D model of the sunflower plant after the pre-processing stage (**A**), segmented leaf-related points (**B**) and segmented stem-related points (**C**).

**Figure 4 sensors-19-01569-f004:**
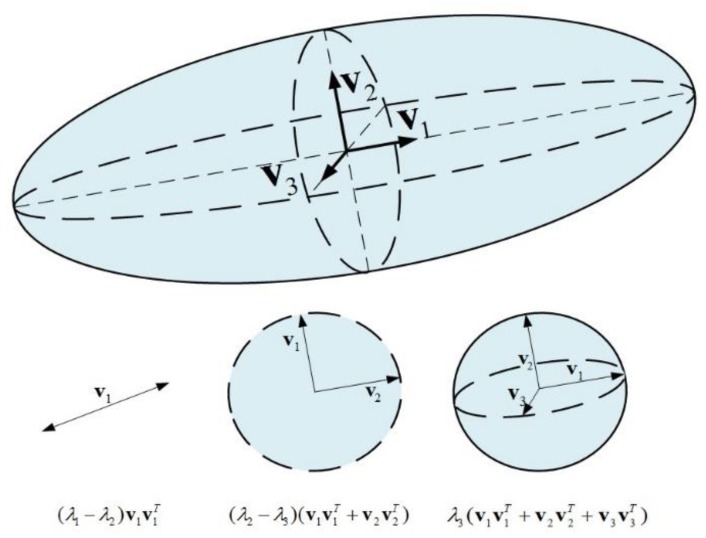
Visualization of a 3-D second-order tensor. The first term in Equation (3) corresponds to a degenerate elongated ellipsoid (linear tensor), with **v**_1_ as its curve normal. The second term corresponds to a circular disc (plate tensor) with **v**_3_ as the surface normal. The third term corresponds to a structure with no preference of orientation (sphere tensor). The stick tensor is given as $(\lambda_{1}-\lambda_{2})v_{1}v_{1}^{T}$, the plate tensor is given as $\left(\lambda_{2}-\lambda_{3})(v_{1}v_{1}^{T}+v_{2}v_{2}^{T}\right)$, and the sphere tensor is given as $\lambda_{3}(v_{1}v_{1}^{T}+v_{2}v_{2}^{T}+v_{3}v_{3}^{T})$.

**Figure 5 sensors-19-01569-f005:**
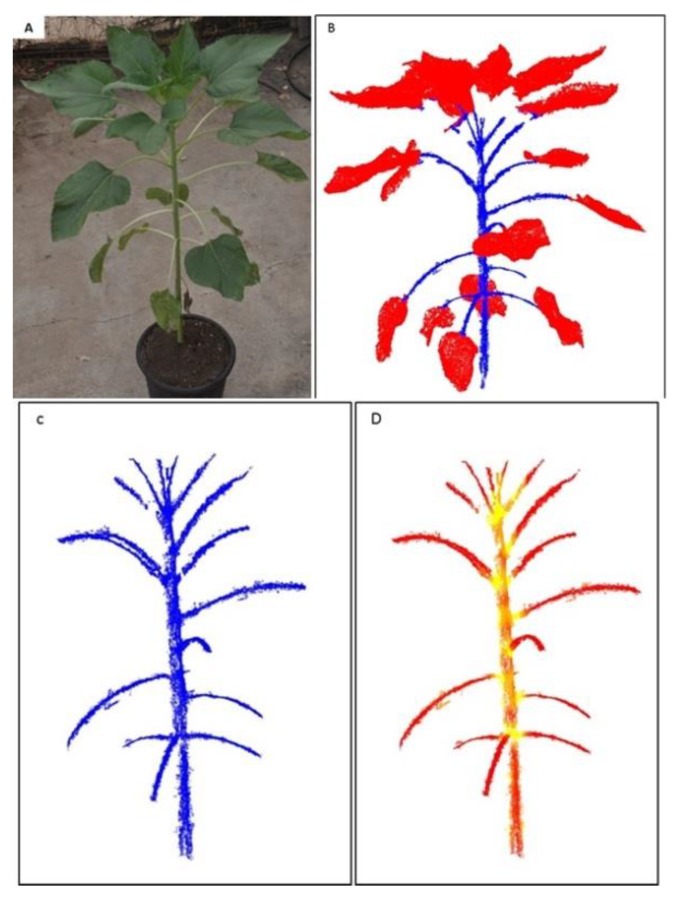
Demonstration of the internode point extraction. Originally acquired image used for the plant 3-D model (**A**) and results of the second-order tensor analysis (**B**). Extraction of the stem-related points (**C**), and how analysis of the *λ*_2_ values allowed detection of the exact location of the nodes location (**D**). Note how the junction area yielded points with high values of *λ*_2_ (yellow), whereas the rest of the stem points were characterized by small values of *λ*_2_ (red).

**Figure 6 sensors-19-01569-f006:**
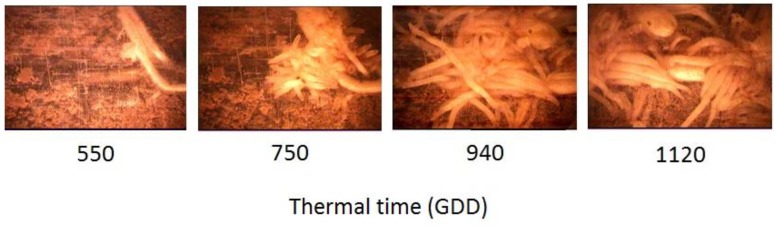
Attachments of *O. cumana* on sunflower plants under controlled conditions, imaged at 10 cm depth by the minirhizotron system, on four consecutive dates, 550, 750, 940 and 1120 GDD after sunflower planting.

**Figure 7 sensors-19-01569-f007:**
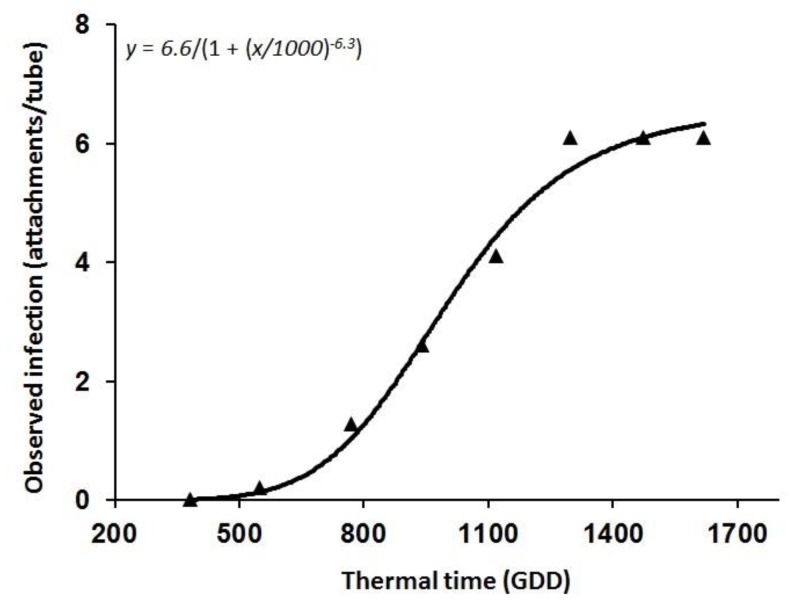
Attachment dynamics (attachments per tube) of *O. cumana* on sunflower plants relative to thermal time measured in GDD.

**Figure 8 sensors-19-01569-f008:**
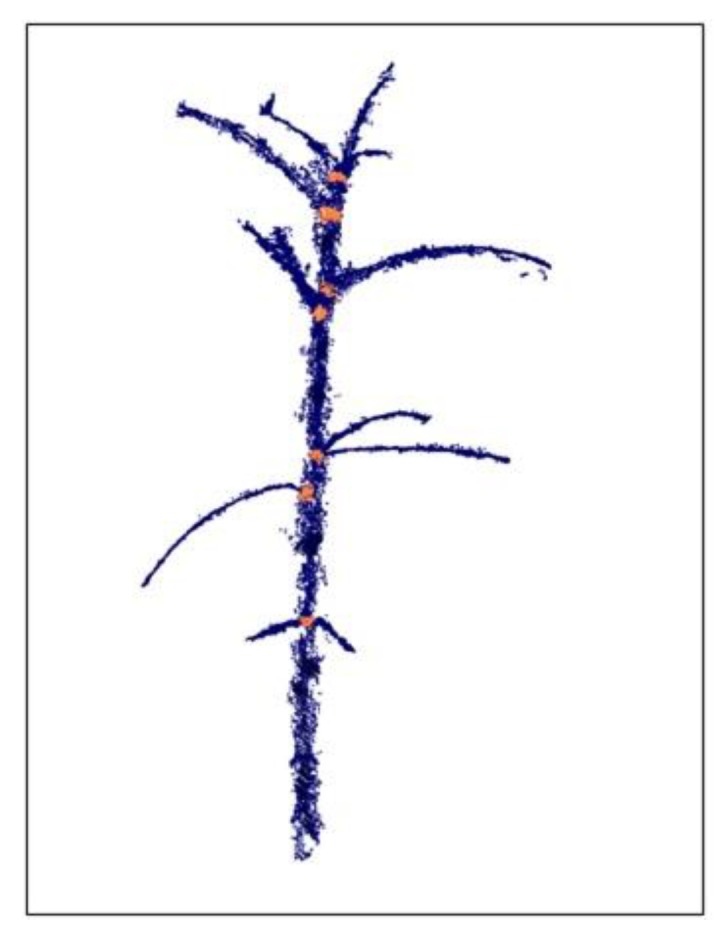
Results of the internode detection algorithm applied on the stem-related points after the segmentation process. Orange dots represent the detected internode points. Setting their position on the stem allowed for estimation of the internode length.

**Figure 9 sensors-19-01569-f009:**
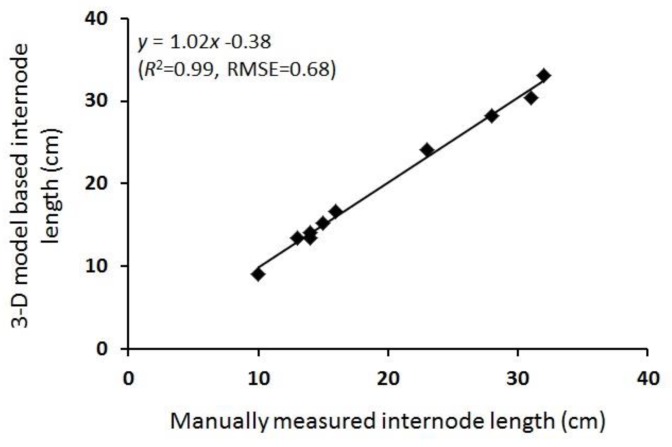
Linear regression between first internode length values extracted from the 3-D model and those measured manually, *y* = 1.02*x* − 0.38 (R^2^ = 0.99, RMSE = 0.68).

**Figure 10 sensors-19-01569-f010:**
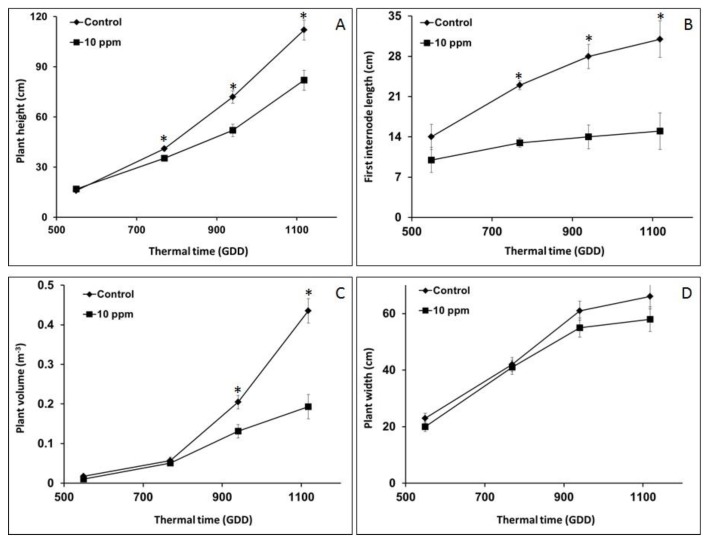
Height (A), first internode length (B), volume (C) and width (D) dynamics of *O. cumana*-infected (■) versus control (♦) sunflower plants. Vertical bars represent standard error of the mean values. * indicates a significant difference as determined by Tukey-HSD test (*n* = 5; *p* ≤ 0.05).

**Table 1 sensors-19-01569-t001:** Attachment dynamics (attachments per tube) evaluated by the minirhizotron system of *O. cumana* on sunflower plants relative to thermal time, measured in GDD.

Parameter	Estimate	SE
*a*	6.62	0.42
*x* _0_	1000.18	34.69
*b*	−6.37	1.17
*R* ^2^	0.98	
*p*	<0.0001	
RMSE	0.32

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
