# Peer review of "3-D Image-Driven Morphological Crop Analysis: A Novel Method for Detection of Sunflower Broomrape Initial Subsoil Parasitism"

_sensors, 2019, doi:10.3390/s19071569_

Round 1
Reviewer 1 Report
This is a well written paper about the use of 3D modelling of a single sunflower plant for the detection of broomrate (O.cumama). The authors suggest that the volumetric features extracted by the 3D model can be used for the early detection of that infection that would lead to minimal use of herbicide and better treatment.
There are several reference errors (e.g., lines 147, 153). This also appears at the reference section where (a) symbols I guess correspond to missing links.
How is the background handled?
As described in 2.4 the model features are compared with manual measurements
In 3.3 the morphological features that can be used to distinguish the infected plants are discussed.
third party software tools are used for the 3D model generation
My main concerns are the following:
1) The 3D model is extracted by snapshots taken around the plant but is this applicable to a real crop? Does each sunflower need to be isolated in a crop? how can this be done? The methodology for a scale-up of the proposed method in a large crop is necessary. Please describe in detail how the proposed method would be used in a real environment
2) the authors admit in the last paragraph of section 4 that other parameters may affect the 3D features they use for the Ocumama detection misleading the diagnosis. Please describe what other methods can be combined to reduce the risk of false diagnosis in real conditions
3) Are there other methods (eg from the referenced approaches) that have been used for the diagnosis of this specific parasite? what is their accuracy? please include a comparison
Author Response
Reviewer #1
Thanks for your useful comments and suggestions.
We have responded to all of your questions and revised the document accordingly. We feel the manuscript is now in better form. Please find a revised version of our manuscript.
1. The 3D model is extracted by snapshots taken around the plant but is this applicable to a real crop? Does each sunflower need to be isolated in a crop? how can this be done? The methodology for a scale-up of the proposed method in a large crop is necessary. Please describe in detail how the proposed method would be used in a real environment
A paragraph with detailed description of the potential application was added (lines 382-397).
2. the authors admit in the last paragraph of section 4 that other parameters may affect the 3D features they use for the Ocumama detection misleading the diagnosis. Please describe what other methods can be combined to reduce the risk of false diagnosis in real conditions
Other methods that can improve our 3D broomrape detection approach were added (lines 404-407).
3. Are there other methods (eg from the referenced approaches) that have been used for the diagnosis of this specific parasite? what is their accuracy? please include a comparison
A comparison about the alternative (hyperspectral) method for broomrape detection was added (lines 339-345).

Reviewer 2 Report
I had the opportunity to review the manuscript "3-D Image-Driven Morphological Crop Analysis: A Novel Method for Detection of Sunflower Broomrape Initial Subsoil Parasitism". The authors demonstrate how evaluation of the crop's 3-D morphology may serve as a proxy for identifying broomrape infection, even before the weed emerges. This study demonstrates the potential value of 3-D modeling for pre-emergence detection of O. cumana, and a phenotyping tool in breeding projects for new broomrape-resistant cultivars. Therefore, the manuscript is within the Scopo of Journal Sensors and has merit to be considered for publication after the authors make the following corrections:
- All citations in the text are wrong. I ask the authors to look at the last manuscripts published in Sensors to correctly format the manuscript;
- I would like to see the study hypothesis, which was well presented in the Introduction, in the Abstract of the paper;
- The authors mentioned between lines 65-69 in the Introduction that "It has been hypothesized that spectral and thermal sensing can be employed to monitor O. cumana establishment, the host plant may be under water stress, which, in turn, may be reflected by differences in absorbance (hyperspectral) patterns or leaf temperatures. " This information is interesting, but readers need more information about it. What physiological mechanisms occur in sunflower that enable the detection of O. cumana from reflectance patterns? Does the host plant increase the respiratory rate and does it modify some wavelengths? These aspects need to be explored better in the Introduction.
- line 135 - correct the number of the equation;
- line 153 - it was not possible to verify the formula inserted in the text. Please correct;
- When reading the manuscript, it was not clear to me where the Tukey test was applied. You may need to insert a caption in the Table or Figure that contains this analysis.
- Include the parameters of the equation in Figure 7 itself;
- Figure 9 - idem to the previous comment;
- The references have been checked and they are ok.
Author Response
Reviewer #2
Thanks for your useful comments and suggestions.
We have responded to all of your questions and revised the document accordingly. We feel the manuscript is now in better form. Please find a revised version of our manuscript.
1. All citations in the text are wrong. I ask the authors to look at the last manuscripts published in Sensors to correctly format the manuscript;
Citations were edited according to the journal style.
2. I would like to see the study hypothesis, which was well presented in the Introduction, in the Abstract of the paper;
As suggested, the hypothesis of the study was added to the Abstract (lines 23-25).
3. The authors mentioned between lines 65-69 in the Introduction that "It has been hypothesized that spectral and thermal sensing can be employed to monitor O. cumana establishment, the host plant may be under water stress, which, in turn, may be reflected by differences in absorbance (hyperspectral) patterns or leaf temperatures. " This information is interesting, but readers need more information about it. What physiological mechanisms occur in sunflower that enable the detection of O. cumana from reflectance patterns? Does the host plant increase the respiratory rate and does it modify some wavelengths? These aspects need to be explored better in the Introduction.
As suggested, more information was added about the physiological aspects associate with broomrape infection and results in possible detection by hyperspectral/fluorescence imaging (lines 83-89).
4. line 135 - correct the number of the equation;
It is correct in the original "Word" file. I guess the conversion to the PDF cause some errors.
5. line 153 - it was not possible to verify the formula inserted in the text. Please correct;
All equation were checked and found correct in the original "Word" file. I guess the conversion to the PDF cause some errors.
6. When reading the manuscript, it was not clear to me where the Tukey test was applied. You may need to insert a caption in the Table or Figure that contains this analysis.
A relevant caption was inserted (into figure 10) mentioning where the Tukey test was applied (lines 631-632). Additionally, the figure was edited accordingly.
7. Include the parameters of the equation in Figure 7 itself;
Image was edited as requested.
8. Figure 9 - idem to the previous comment;
Image was edited as requested.
Round 2
Reviewer 1 Report
The line numbering of the responses to my comments is wrong. Thus I cannot find what are the authros responses to my comments. I can only suspect that:
Their response to my comment about
"The 3D model is extracted by snapshots taken around the plant but is this applicable to a real crop? Does each sunflower need to be isolated in a crop? how can this be done? The methodology for a scale-up of the proposed method in a large crop is necessary. Please describe in detail how the proposed method would be used in a real environment"
is given in lines 362-374 where the authors state that
"Our 3-D modelling approach is based on standard consumer cameras and can be applied over large fields with potential for complete automated data extraction and interpretation. Such models will probably be most effective with single-sided images acquired from a close range, with cameras mounted on tractors and other mobile platforms."
I still cannot imagine how feasible will be to use tractors or other vehicles riding around a single plant in order to take pictures and form the 3D model. A tractor could take a route within a field and take several photos displaying different plants not 360o of the same plant. It would be more realistic if the authors said that the farmer can isolate a single plant take surrounding photographs and try to extrapolate the results of the single plant to the whole field with additional information as said in lines 380-383 (are they part of a response to a different comment of mine?)
I asked some comparison with other references. I found in lines 325-329 some references to analyses in different wavelengths but this is not sufficient. It would be useful if the success rate was given of different diagnosis methods for the same disease even if they are totally different than the 3D model used here. If there aren't other quantitive methods to compare concerning the same disease at least compare similar 3D models for the diagnosis of other diseases.
There are still formatting errors. The reference at line 159 is not displayed appropriately. It seems that the revision was not prepared carefully and the benefits of the proposed approach are not obvious.
--------------------------------------------------------------
I suspected that the responses to my comments were the ones in the lines indicated in the second file you sent me and I was right.
I see that I cannot modify my review in the online system but what
would change would be the Reject option to Major Revision since the
reasons I describe in my review are still valid and I am not satisfied
by the modifications performed by the authors. So if you like you can
leave my comments as they are but change my suggestion to Major Revision
and give the authors one more chance to extend their paper, adding more
comparison with other approaches and explaining better how their system
would be used.
Author Response
Thanks for your review. Please see responses in attachment.

Round 3
Reviewer 1 Report
I admit that the contribution of the proposed method is significant. Although I insist that it will be difficult to apply this method in the field and my request for comparison with other approaches was not satisfied, I suggest to accept this paper since the other reviewers did not insist in such issues